# The Extraordinary Variety and Complexity of Minerals in a Single Keokuk Geode from the Lower Warsaw Formation, Hamilton, Illinois, USA

**Nova Mahaffey *** and **Robert B. Finkelman** 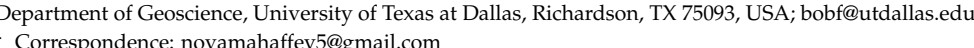

Department of Geoscience, University of Texas at Dallas, Richardson, TX 75093, USA; bobf@utdallas.edu
*   Correspondence: novamahaffey5@gmail.com

**Abstract:** We performed an extensive optical and chemical analysis of a single Keokuk geode using electron microscopy (SEM) with energy dispersive X-ray (EDX) spectroscopy that revealed an extraordinary array of minerals and multiple, complex cycles of mineralization. We identified at least 15 minerals including 5 that, to our knowledge, have not been reported in previous studies of these geodes. Along with bitumen we have described the occurrence of REE's, and other unidentified phases containing metals such as chromium, nickel, molybdenum, tin, copper, zinc, and lead. Additionally, preliminary thin-section analysis reveals the occurrence of the tentatively identified minerals zircon, rutile, and xenotime as well as grains containing gold and silver within the chalcedony shell. The presence of these potentially economically valuable minerals warrants further investigation into the micro-minerology of Keokuk geodes. Our SEM/EDX analysis reveals an array of complex mineral assemblages, intergrowths, and inclusions that help chronologically link multiple stages of paragenesis occurring in different locations within the geode. Consequently, morphology and intricate microstructures provide a window into the extreme complexity of mineral crystallization. The majority of micro-minerals we have observed correspond with the later stages of geode paragenesis, thus providing a detailed record of the secondary mineralization processes which occurred over thousands to millions of years.

**Keywords:** geodes; micromineralogy; paragenesis; Keokuk region; crystal shape; mineral association; scanning electron microscopy

## 1. Introduction

Initially this project began as an undergraduate research project (by NM) on the mineralogy of geodes from the famous Keokuk, Iowa area but quickly evolved into a detailed characterization of a single geode. We randomly selected this geode to be the first in a mineralogic study of a full suite of Keokuk geodes, however we ended up spending virtually all of our time characterizing the mineralogy of this single geode because of its extraordinary variety and complexity. To our knowledge there was only one other study [1] that focused on the mineralogy of a single geode, a large (approximately 43 × 23 cm, 23 kg) amethyst geode from Brazil containing five minerals. Since the first report on Keokuk geodes [2] was published, a multitude of other studies have described the mineralogy of these Midwestern geodes (see, for example [3–9]). However, we believe this study provides the most intensive mineralogical analysis of a single Keokuk geode, describing a total of 15 minerals, 5 of which are unique to this study. Additionally, we have observed various unidentified phases containing heavy metals and REE's as well as bitumen and fluid inclusions. The majority of the micro-minerals we observed correspond with the later stages of geode paragenesis, thus providing a detailed record of the secondary mineralization processes in the geode that occurred over thousands to millions of years. We will illustrate the most comprehensive paragenesis of a single geode based on the minerals

described below and conclude with a few comments about the possible significance of these observations and suggestions for future investigations.

## 2. Background

The book *Geodes* describes the Keokuk region as the "best known locality in the America's for sedimentary geodes ... " [5]. These geodes are found exclusively in the dolomitic beds of the Warsaw and Keokuk Formations in the Midwestern region of the United States, around a 60-mile radius from Keokuk, Iowa (Figure 1) [5]. These formations are sedimentary marine deposits which are the consequence of a regressing epicontinental sea during the Mississippian period about 340 million years ago. In 2018, fifty geodes were collected from an area about 10 feet long by 3 feet high at Jacobs Geode Farm in Hamilton, Illinois just across the Mississippi River from Keokuk, (Figure 1). This particular site corresponds to the Lower Warsaw Formation, for more details about the geology, explanation of origin, and a comprehensive list of minerals can be found in the "Field trip guide to kaolinite geodes in Hamilton, Illinois, USA" [3].

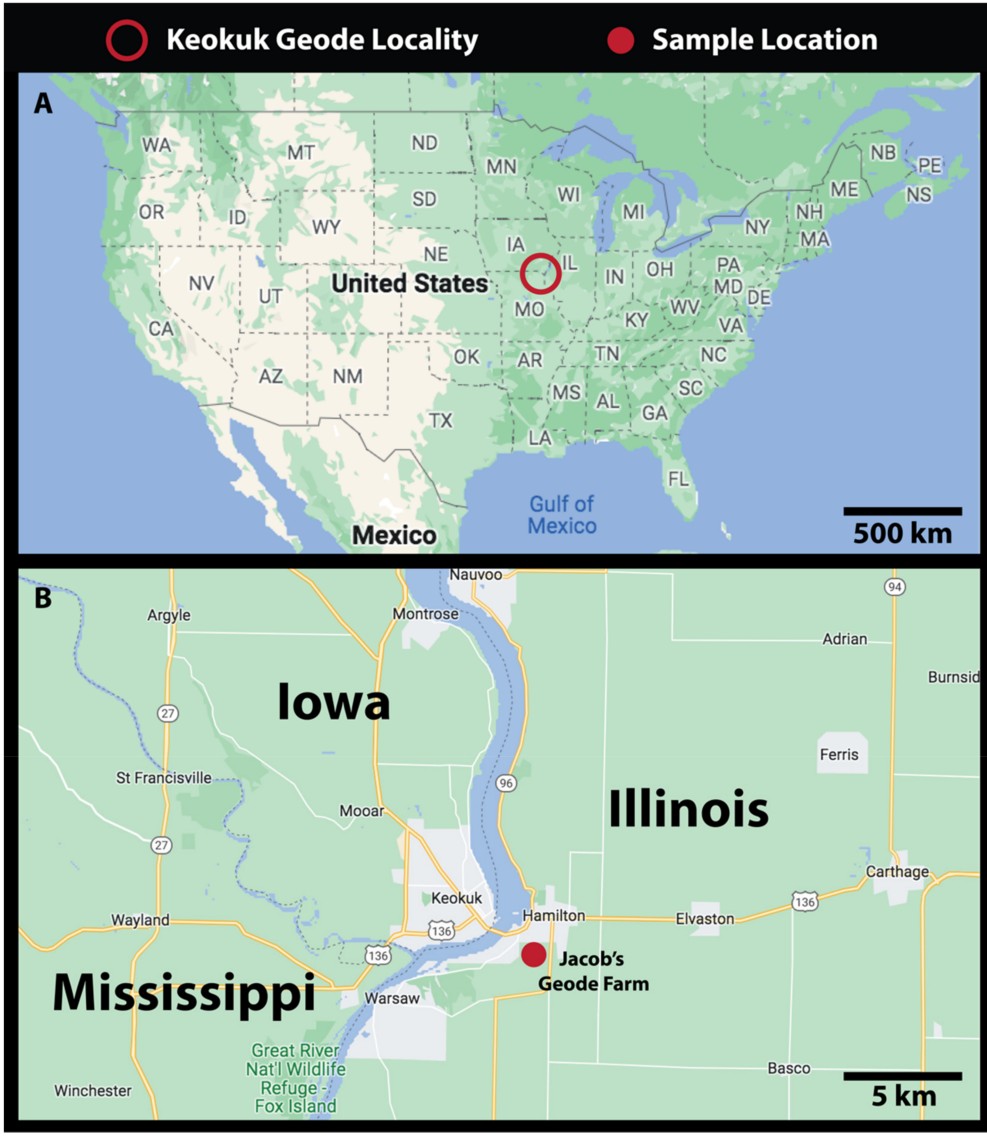

**Figure 1.** Keokuk geode region and collection site map. (**A**) A map of the united states indicating the general Keokuk geode locality. (**B**) A map the tri-state region indicating where the collection site Jacob's Geode Farm is located.

### 3. Methods

From the suite of geodes, about a half dozen were randomly selected and cut in half with a rotary saw with a diamond tipped blade. On initial observation we noted the characteristic chalcedony shell with inwardly radiating druzy quartz crystals and patches of a white "dust" we assumed to be kaolinite. Some of the geodes featured distinct brown and pale-yellow rhombic crystals, remnants of red-brown oxidation, and black organic particulate. To begin our study, we selected one geode (approximately 8.5 cm by 5 cm and weighing 324.2 g) that appeared to be quite interesting exhibiting a "double geode" feature with two distinct cavities (Figure 2). One half of the geode was broken into smaller fragments that were examined with a binocular microscope. Following observation and photography with a binocular microscope selected fragments of the geode were mounted and lightly carbon coated for examination in a scanning electron microscope with an energy dispersive X-ray detector (SEM/EDX) at the University of Texas at Dallas using a JEOLJSM-IT100 SEM/EDS. The collection parameters were an accelerating voltage of 15–20 kV with a probe current of 50–65 pA, using back-scatter electron detection in low-vacuum mode. We also conducted a preliminary polarized light microscopy and SEM/EDX analysis on a thin-section cut from the geode (Figure 3). Mineral identification is based on the most common mineral associated with the chemistry. Unfortunately, most of the accessory minerals were just a few micrometers in size, too small to isolate and far too small to obtain X-ray diffraction patterns from.

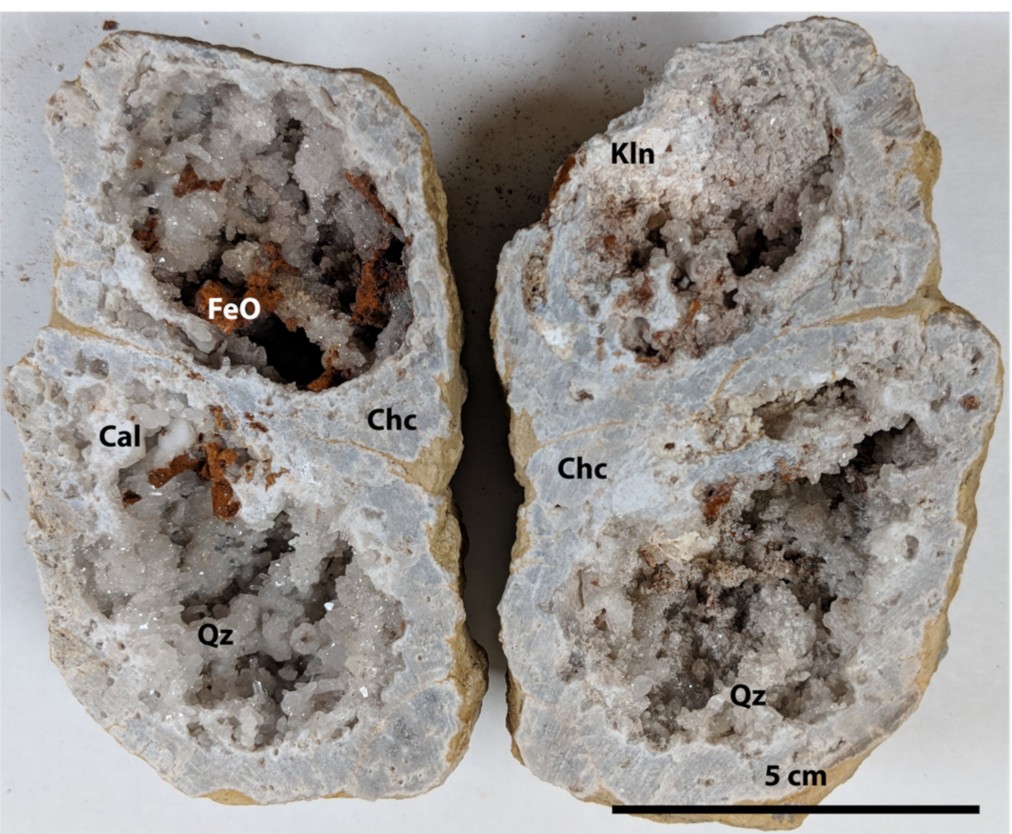

**Figure 2.** Keokuk geode halves exhibiting a "double geode" feature with two distinct cavities. Note the chalcedony (Chc) shell with inwardly radiating "druzy" quartz (Qz) crystals, rhombic calcite (Cal) crystals, patches of a white kaolinite (Kln) "dust", and what appears to be remnants of iron-oxide (FeO).

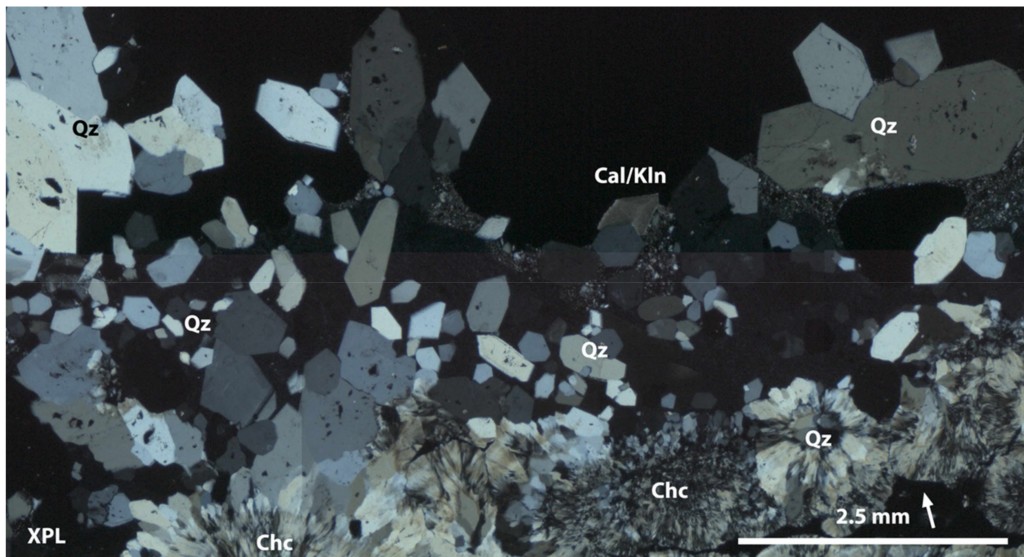

**Figure 3.** Thin-section of a portion of the geode's shell shows pseudo-fibrous chalcedony (Chc) with quartz (Qz) terminations, larger discrete quartz (Qz) "druzy" crystals, and a rhombohedral calcite-kaolinite crystal (Cal/Kln). The arrow indicates the direction towards the center of the cavity. (TS, XPL).

## 4. Results

### 4.1. Mineralogy

We describe the minerals encountered in this geode in approximate order of abundance. Those minerals in are believed to be new to Keokuk geodes. Identifications of the minerals are based primarily on the chemistry and, where feasible, the crystal form.

### 4.1.1. Chalcedony—SiO₂

The chalcedony shell constitutes the largest volume of the geode. In thin-section under polarized light the chalcedony exhibits a pseudo-fibrous habit with growth lines in a chevron pattern (Figures 3 and 4). Splays of chalcedony surround micron sized quartz grains and doubly terminated crystals which forms the matrix of the shell (Figure 4). This is consistent with previous descriptions of the chalcedony shell as a mosaic of quartzine, a pseudo-fibrous variety of chalcedony [10]. The chalcedony shell contains an abundance of inclusions including siderite, pyrite, and probable baryte, and celestine.

### 4.1.2. Quartz—SiO₂

Quartz occurs as clusters of micron-sized subhedral grains and individually doubly terminated crystals within the chalcedony shell (Figure 4). Small (less than 0.5 mm) quartz terminations project radially from the chalcedony (Figure 5) while discrete, larger (around 1–3 mm) crystals cover the internal surfaces of the shell creating the characteristic "druzy" appearance (Figures 2 and 3). The larger of the quartz crystals contain inclusions of pyrite and some even feature fluid inclusions. These druzy quartz crystals are also associated with growths of siderite, ferroan rhodochrosite, rhodochrosite, calcite, baryte and hollandite. In one crystal the opaque white color of the quartz was due to a coating of micron-sized silica spherules. Kaolinite incorporated in the surface of quartz interferes with the crystal faces giving the crystal a cloudy appearance (Figure 6) while quartz associated with siderite shows red-brown localized iron staining.

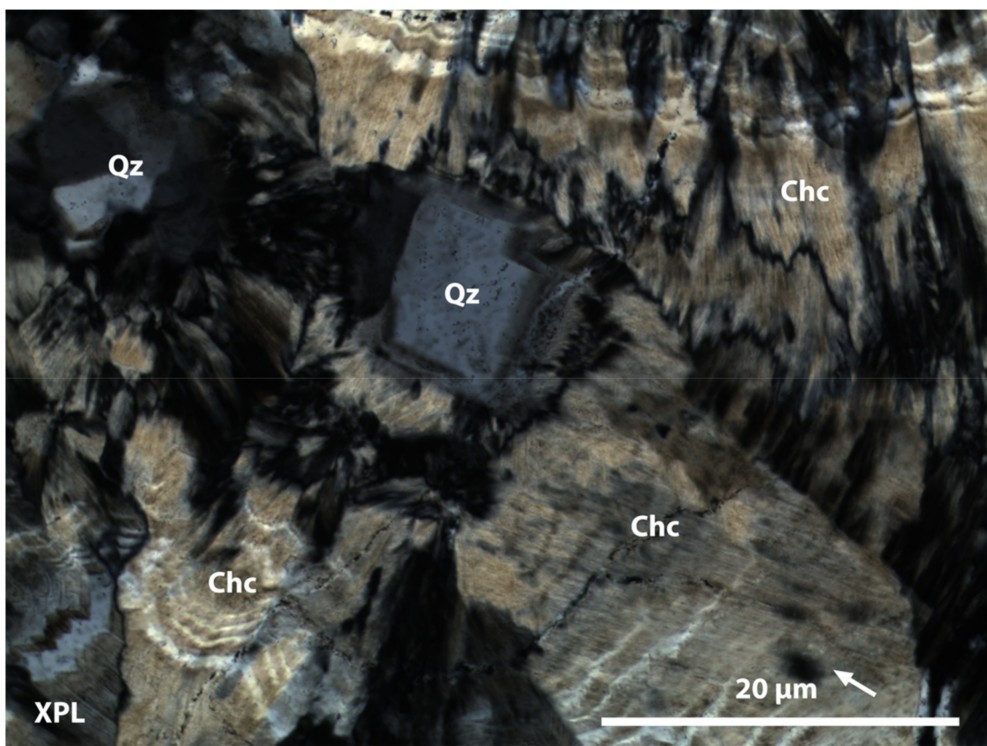

**Figure 4.** Thin-section reveals splays of pseudo-fibrous chalcedony (Chc) surrounding a doubly terminated quartz (Qz) crystal and exhibiting growth lines in a chevron pattern. The arrow indicates the direction towards the center of the cavity. (XPL).

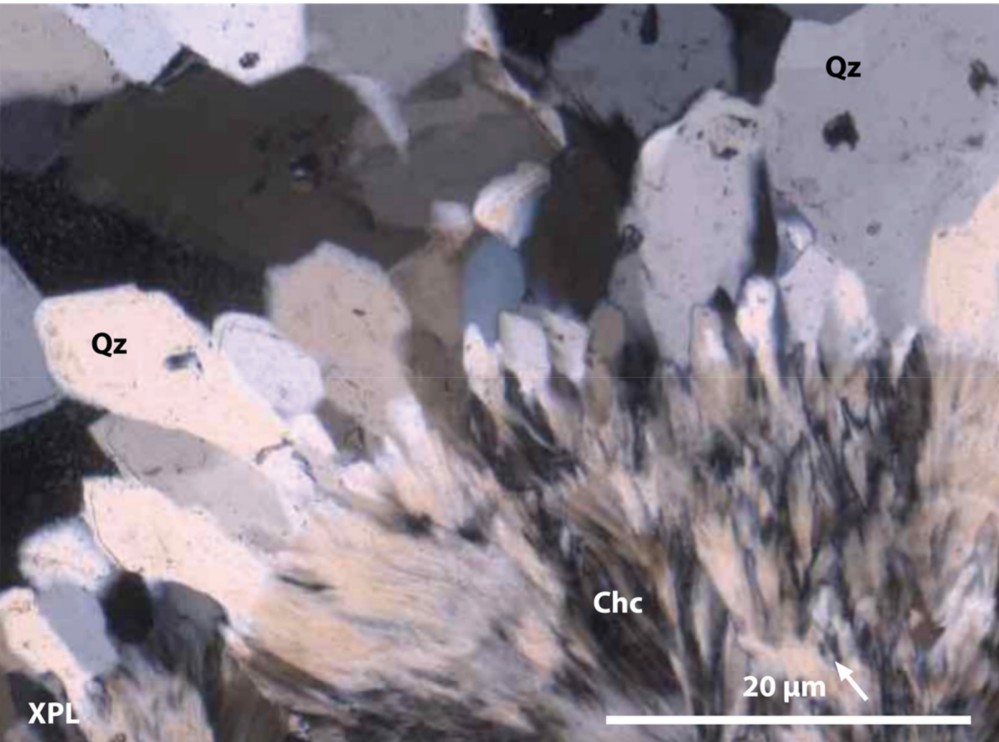

**Figure 5.** Thin-section reveals that quartz (Qz) terminations project directly from chalcedony (Chc). The arrow indicates the direction towards the center of the cavity. (XPL).

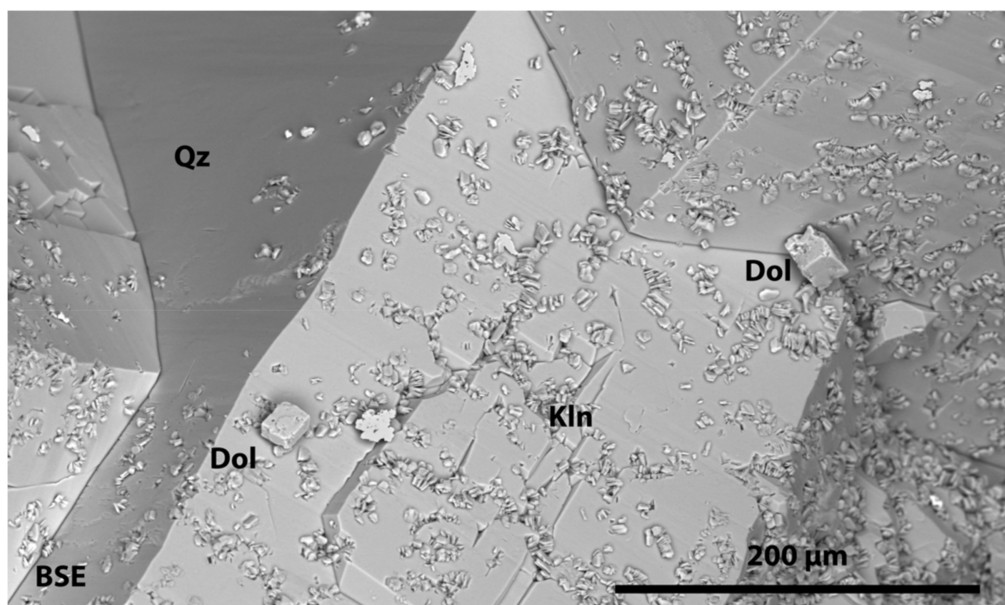

**Figure 6.** Kaolinite (Kln) crystals incorporated into the surface of a quartz (Qz) crystal accompanied by two rhombohedral dolomite (Dol) crystals. Note: The brighter spots are remnants of iron-oxide. (BSE).

### 4.1.3. Calcite—CaCO₃

What we describe as calcite has varying concentrations of magnesium but not high enough to be considered dolomite. Calcite occurs in four distinct habits in the geode:

1.  Calcite Rhombohedra—Calcite forms rhombohedral macro-crystals (about 1–5 mm) some of which exhibit a semi-saddle-like habit and form among the druzy quartz. The clusters of brown crystals we initially observed are identified as calcite with a siderite coating (Figure 7).

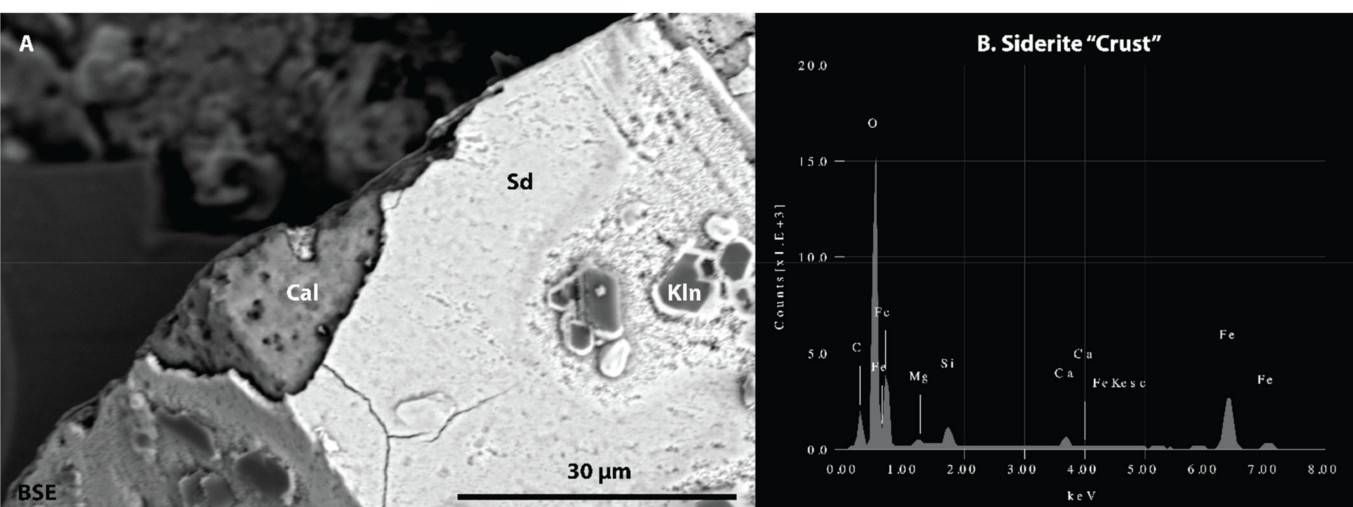

**Figure 7.** (**A**) Siderite "crust" (Sd) coating a calcite (Cal) crystal with kaolinite (Kln). (BSE) (**B**) EDX spectra of the siderite "Crust" (Sd).

2.  Calcite "Flow"—Calcite containing up to about 15% kaolinite was observed on top of and in-between quartz crystals, and other mineral assemblages. This habit is characterized by ripple-like growth lines that indicate directionality (Figure 8A). In some instances, the "flow" builds into a stair-stepped structure with pits that contain kaolinite (Figure 8B).

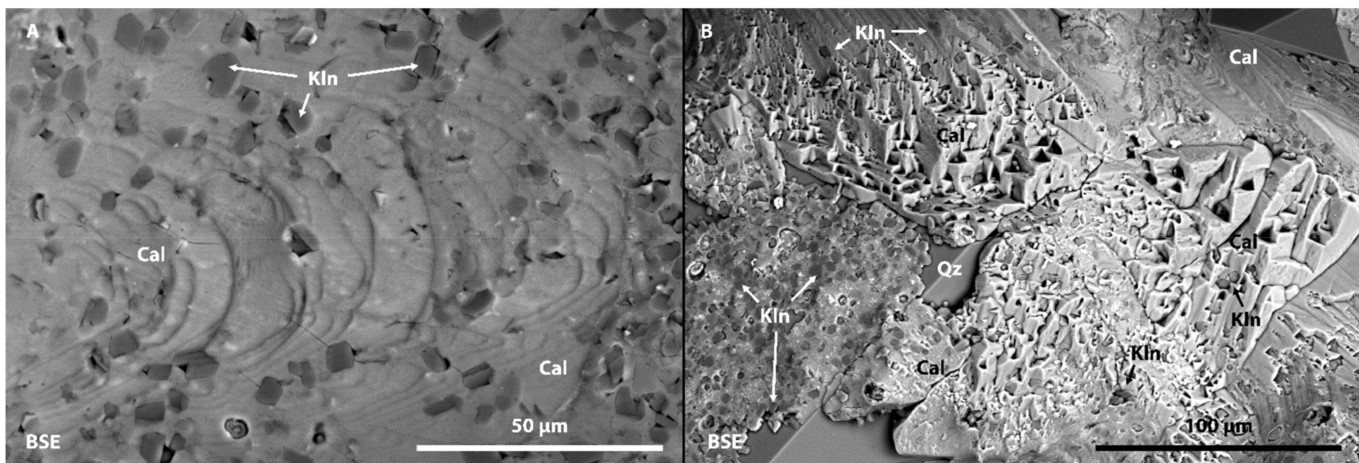

**Figure 8.** Calcite "flow" (Cal) incorporating kaolinite (Kln) crystals and (**A**) exhibiting growth lines that mimic ripple marks. (**B**) exhibiting a stair-stepped box-like structure. (BSE).

3.  Calcite "Fill"—A particular conglomerate of about 20%–50% kaolinite and 5% dolomite in a calcite matrix is observed "filling" voids between quartz crystals, sometimes incorporating pyrite/goethite crystals. A similar calcite "fill" is associated with golf-ball-like siderite growths, baryte, and fragments of feldspar (Figure 9).

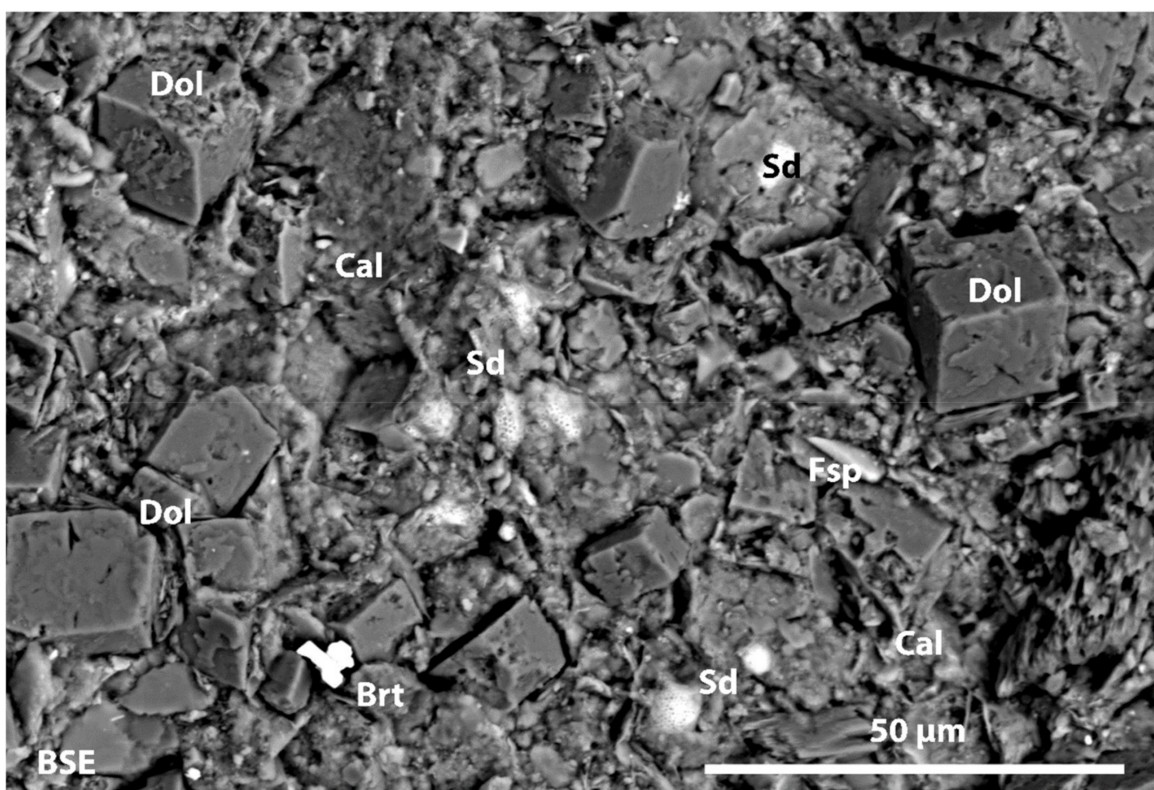

**Figure 9.** Calcite "fill" (Cal) incorporating rhombohedral dolomite (Dol) crystals, fragments of feldspar (Fsp), golf-ball like growths of siderite (Sd), with a set of baryte (Brt) crystals. (BSE).

4.  Calcite "Euhedra"—Opaque white to pale yellow rounded rhombohedral crystals of calcite are observed among the quartz crystals. From the thin-section it was apparent that kaolinite is incorporated through the entirety of the crystal, making up about 50% of their volume (Figure 2). Kaolinite at the surface of the crystal creates a pitted

texture, likely responsible for the crystal's dull appearance. These crystals are also found within voids in the chalcedony shell.

### 4.1.4. Siderite—FeCO$_3$ and Rhodochrosite—MnCO$_3$

Members of the siderite-rhodochrosite series are reflected in the habits described below. While we note the presence of calcium, as previously mentioned, identification is based on the most common minerals associated with the chemistry. For Simplification, we describe ferroan rhodochrosite as a ratio of near equal parts iron to manganese. Thus, siderite will be considered as containing more iron while rhodochrosite more manganese (Figure 10). The ferroan-rich rhodochrosite observed may be the ponite variant [11].

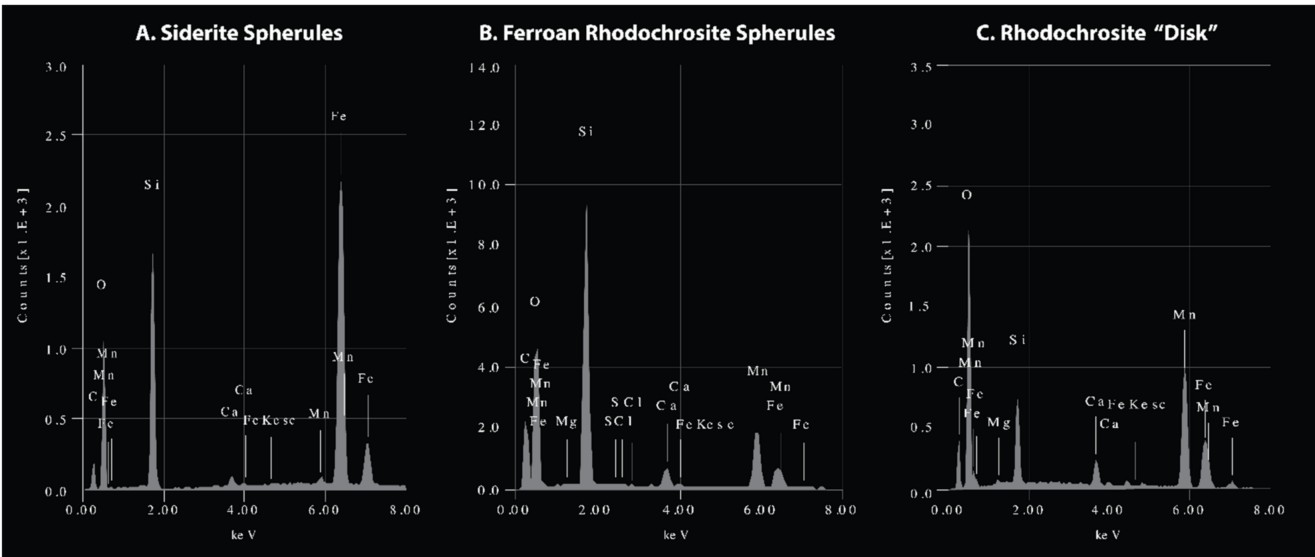

**Figure 10.** EDX spectra for (**A**) Siderite spherule (**B**) Ferroan Rhodochrosite Spherule (**C**) Rhodochrosite "disk.

1.  Siderite "Crust"—A micron thin coating of siderite is seen coating calcite crystals corresponding to the tan to brown color of the rhombohedral crystals observed among the druzy quartz (Figure 7). In a previous study of Keokuk geodes [12] the comparable "brown iridescence" observed was reported as stilpnosiderite, a calcium-bearing variety of limonite.
2.  Siderite "Framework"- Siderite forms within what appear to be cleavage cracks in siderite-crusted calcite crystals creating an internal lattice-like structure (Figure 11). The subsequent dissolution of calcite exposes this structure which we refer to as the siderite "framework" (Figure 12). The "framework" reflects the shape of the original rhombohedral calcite (Figure 12) crystal and exhibits botryoidal textured walls that incorporate hollow siderite spherules. Oxidation of the "framework" is responsible for the remnants of red-brown oxidation initially observed under the binocular microscope (Figure 2).
3.  Siderite "Nanoneedles"—Sub-micron acicular siderite crystals are associated with kaolinite crystals found associated with the siderite "crust" (Figure 7) and the siderite "framework" (Figure 13). Pom-pom-like clusters of these siderite "nanoneedles" appear almost exclusively along the edges of the kaolinite crystals, leaving the hexagonal faces exposed (Figure 13).

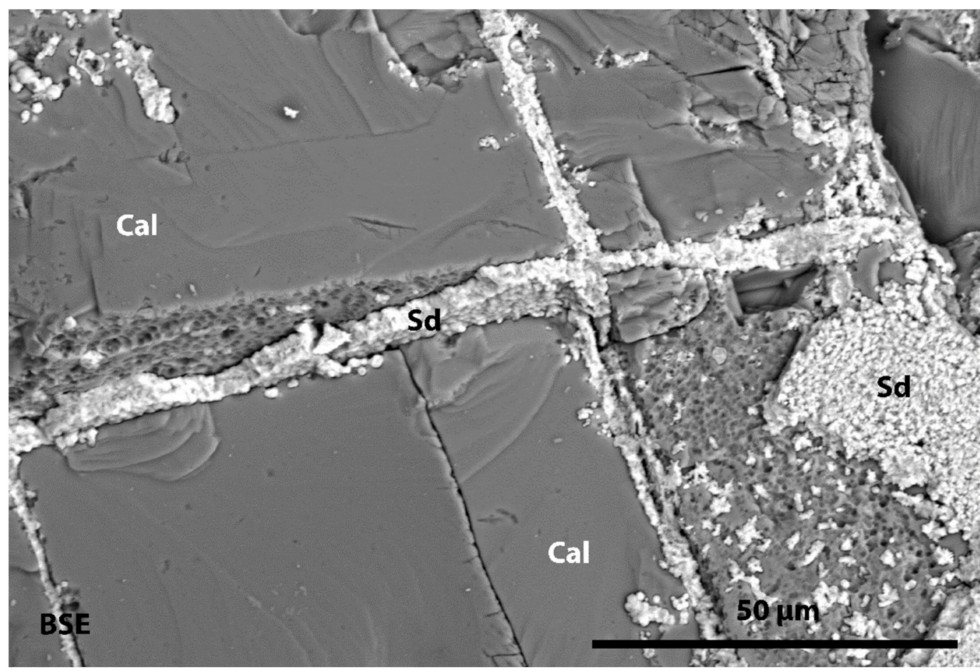

**Figure 11.** The surface of a calcite (Cal) crystal shows siderite (Sd) in cleavage cracks. (BSE).

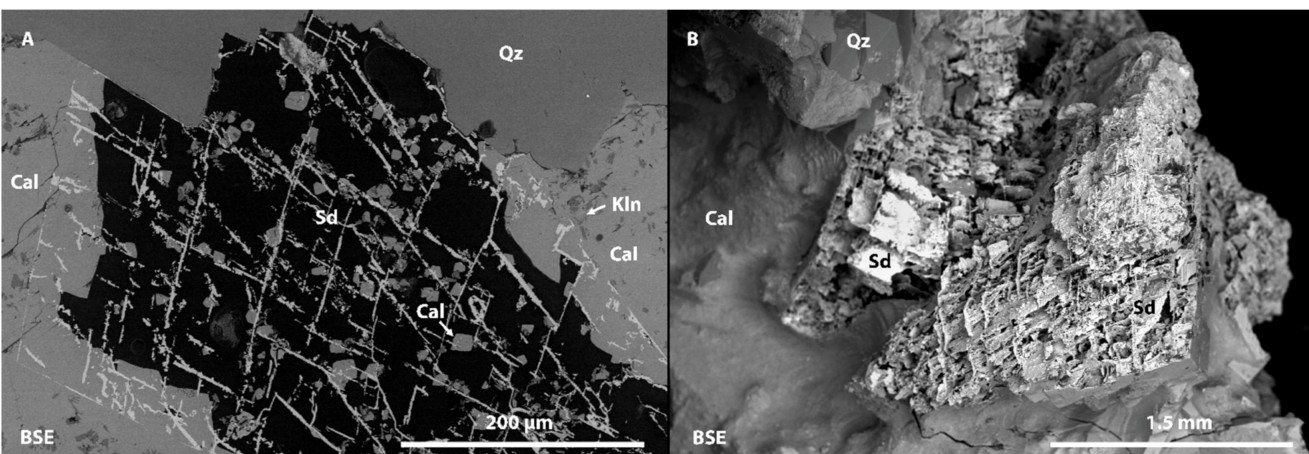

**Figure 12.** (**A**) Thin-section reveals that the siderite "framework" (Sd) follows the cleavage planes within the original calcite (Cal) crystal. (BSE). (**B**) The siderite "framework" (Sd) reflects the original rhombic habit of the calcite (Cal) crystal. (BSE).

4.   Spherules—Micron-sized spherules were observed in groups of about 3–5 on top of druzy quartz terminations (Figure 14). From group to group the spherules exhibit varying concentrations of iron and manganese, corresponding with siderite through ferroan rhodochrosite (Figure 10A,B). When associated with the "framework" larger (around 10 microns in diameter), hollow siderite spherules exhibit growth rings with slightly varying manganese concentrations (Figure 15). Similar sized spherules associated with the calcite-kaolinite "fill" exhibit a pitted golf-ball like texture (Figure 8) that the thin-section reveals is the result of a conglomeration of micron-sized sub-spherules.

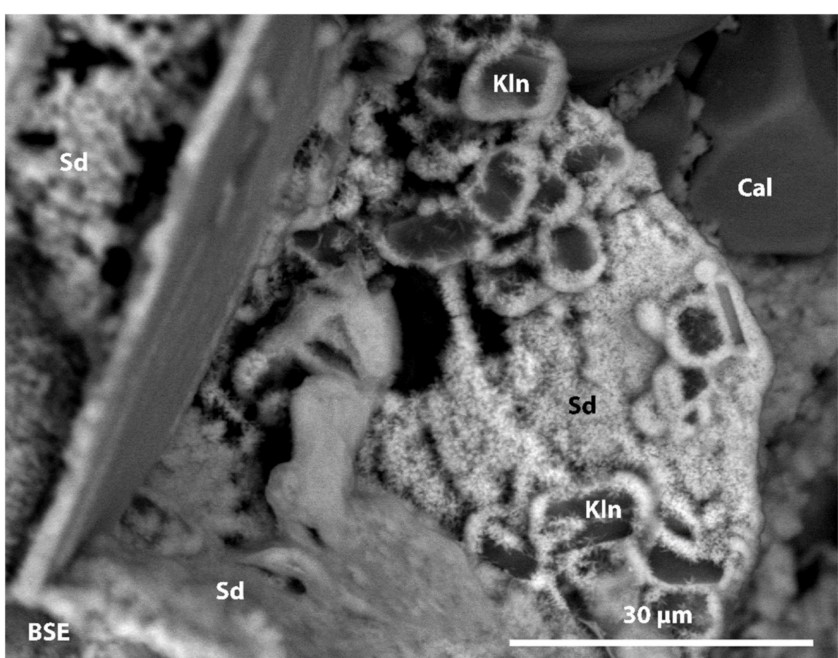

**Figure 13.** Sub-micron acicular siderite (Sd) "nanoneedle" crystals form "pom-pom" like clusters on the edges of kaolinite (Kln) crystals that have settled on top of a piece of the siderite "framework" (Sd). (BSE).

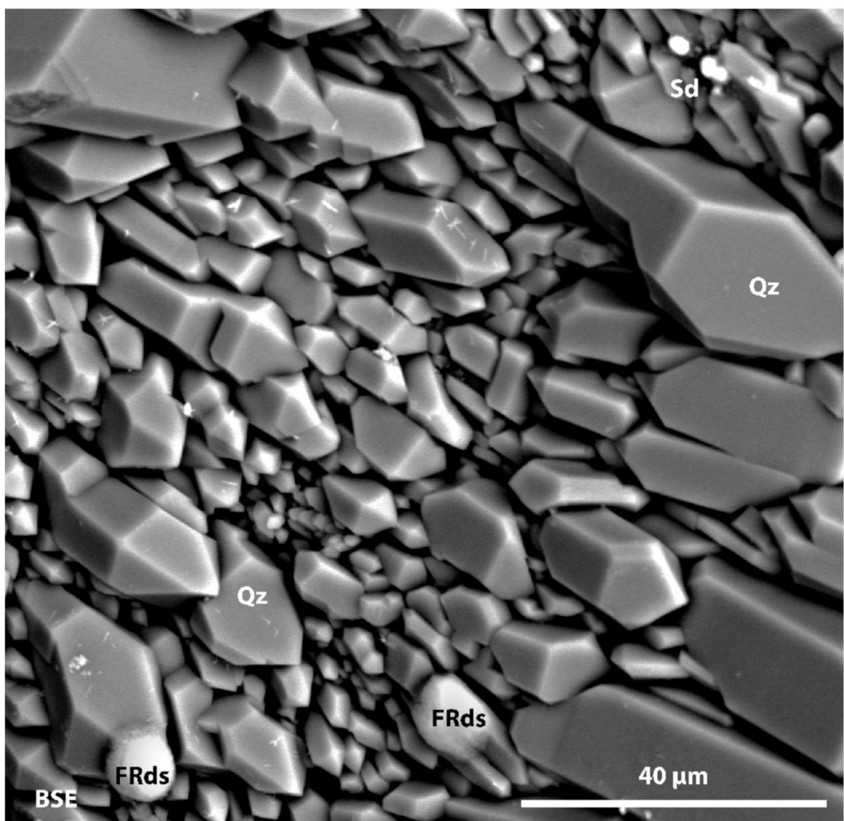

**Figure 14.** A pair of ferroan rhodochrosite (FRds) spherules near a group of siderite (Sd) spherules on top of quartz (Qz) terminations. (BSE).

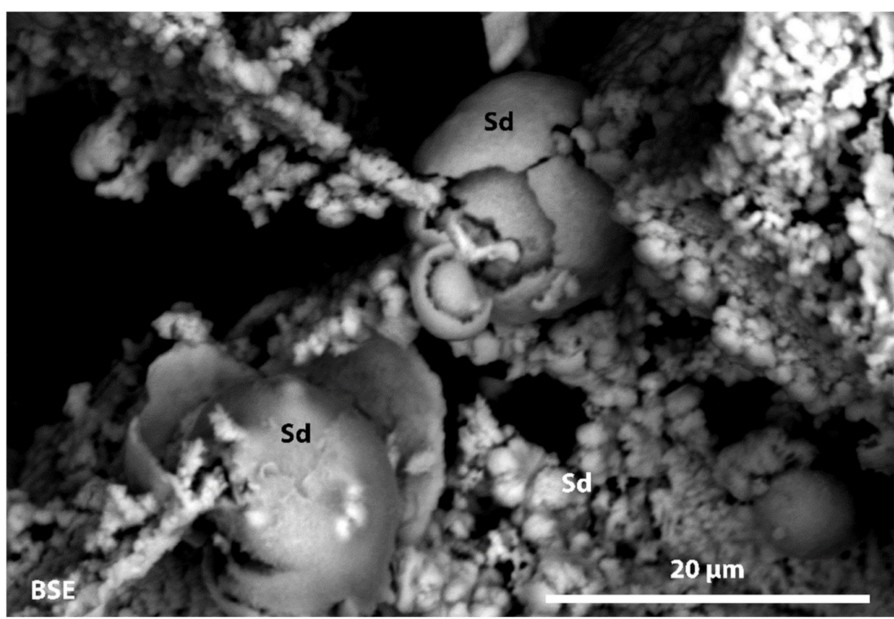

**Figure 15.** Siderite (Sd) spherules within the siderite "framework" exhibit growth rims. (BSE).

5.  Rhodochrosite "Disks"- Rhodochrosite, with comparatively less iron than the spherules (Figure 10C), exhibits a flat circular habit (Figure 16). These beveled "disks" averaging 20–50 microns in diameter and are observed on top of larger druzy quartz crystals (Figure 16).

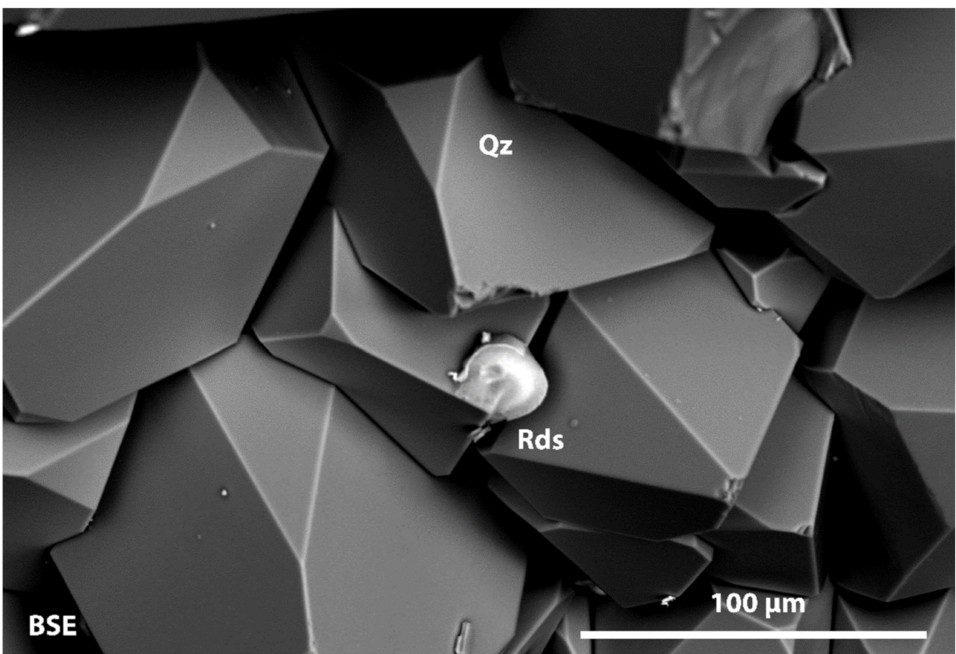

**Figure 16.** A beveled rhodochrosite "disk" (Rds) on top of a quartz (Qz) crystal. (BSE).

### 4.1.5. Kaolinite—$Al_2Si_2O_5(OH)_4$

The white "dust" observed throughout the geode is confirmed by SEM/EDX as probable kaolinite (Figure 2). Kaolinite is easily identified by its platey hexagonal clay-sized crystals and is associated with chalcedony, quartz, calcite, and siderite (Figures 6–8 and 13).

### 4.1.6. Pyrite—FeS$_2$

Pyrite forms perfect euhedral crystals including cubes and octahedrons ranging from 10–100 microns in size (Figure 17). These micro-crystals occur as common inclusions in druzy quartz crystals and in the chalcedony shell. Under the binocular microscope, pyrite crystals exhibit a dark iridescence from the surficial oxidation to goethite. In thin-section, similar pyrite crystals exhibiting a rim of goethite are found within the calcite-kaolinite "fill" (Figure 18).

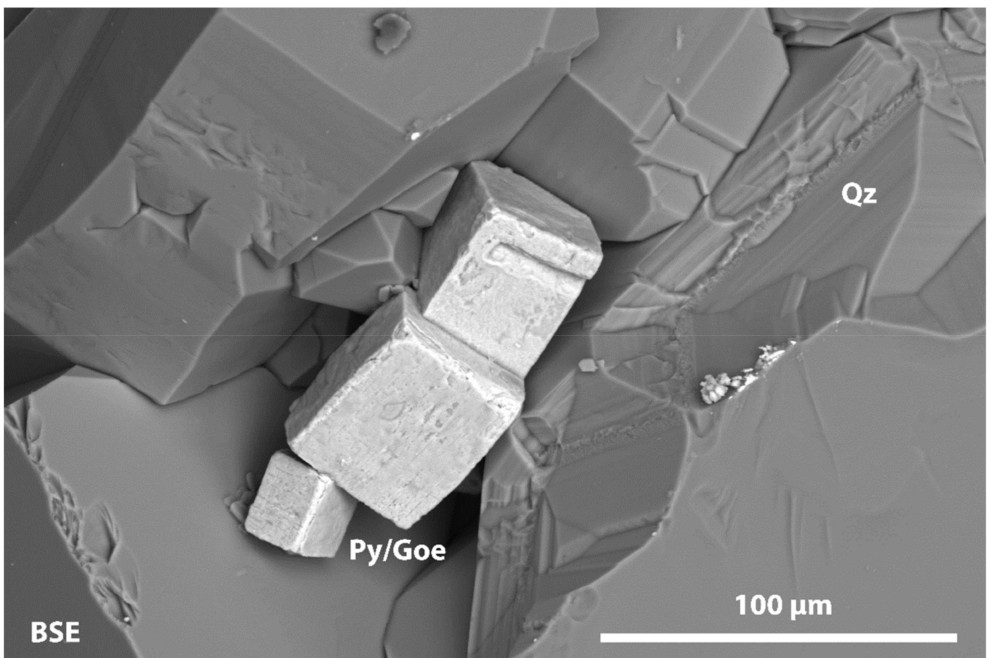

**Figure 17.** Pyrite (Py) micro-cubes nestled among a cluster of quartz (Qz) crystals exhibit a hatch-like texture from surficial oxidation forming goethite (Goe). (BSE).

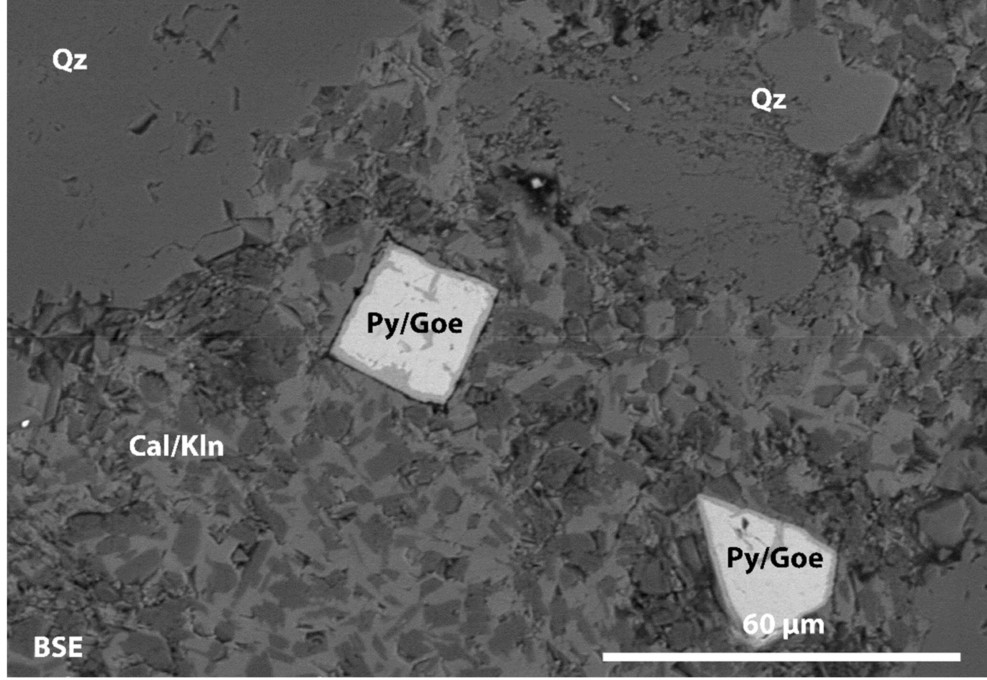

**Figure 18.** Thin-section of the calcite "fill" including kaolinite (Cal/Kln) and several pyrite (Py) crystals that exhibit a rim of goethite (Goe). (BSE).

### 4.1.7. Goethite—FeO(OH)

Goethite, resulting from the oxidation of pyrite, exhibits a dark grey-blue iridescence as observed with a binocular microscope. Under the SEM, goethite features perpendicular striations that create a hatch-like texture on the surface of the pyrite micro-crystals (Figure 17). The thin-section reveals that goethite forms a micron thick rim around similar pyrite crystals found within the calcite-kaolinite "fill" (Figure 18).

### 4.1.8. Dolomite—CaMg(CO$_3$)$_2$

Rhombohedral dolomite crystals around 10 to 20 microns in size were found incorporated in the calcite-kaolinite "fill" (Figure 9) and on top of a kaolinite-included quartz crystals (Figure 6).

### 4.1.9. Feldspar Group

What appear to be fragments of the feldspar group are found incorporated in the calcite-kaolinite "fill" (Figures 9 and 19). These grains exhibit various concentrations of potassium, calcium, and sodium.

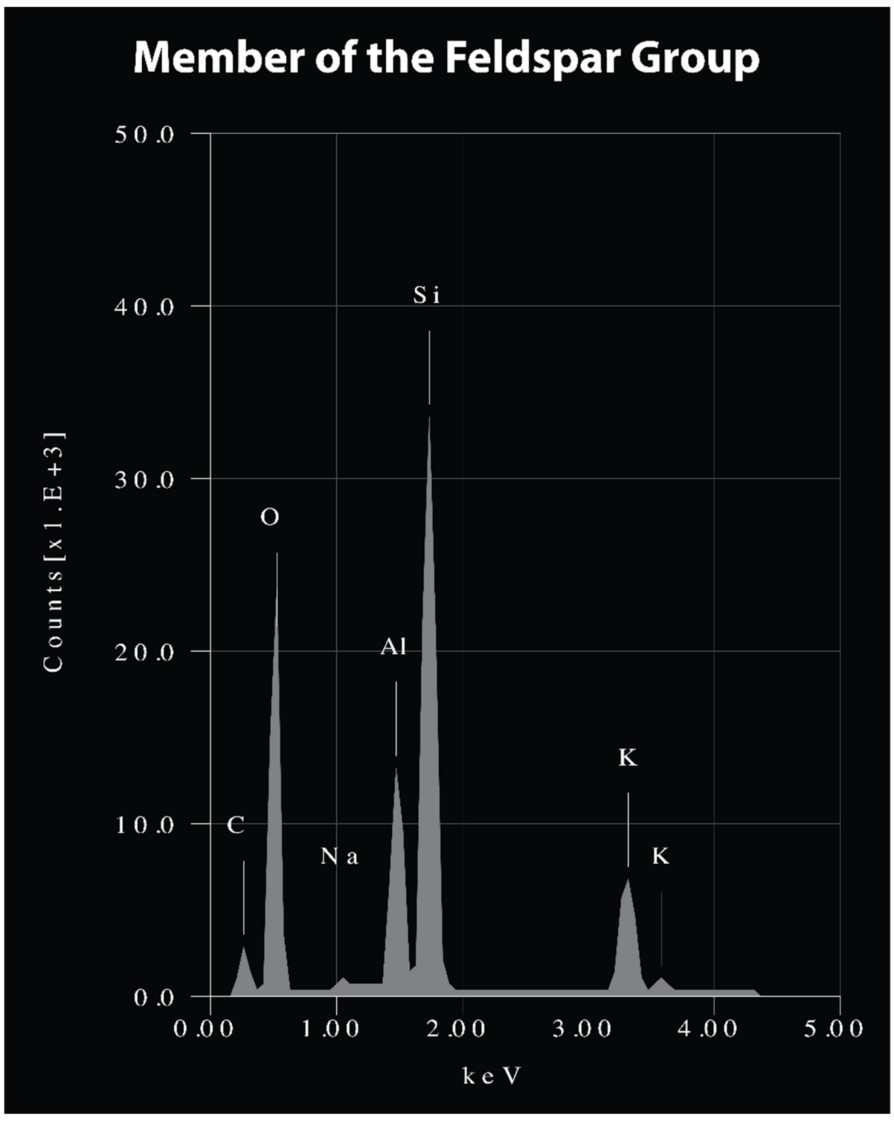

**Figure 19.** EDX spectra of feldspar, reference Figure 9 for an example.

### 4.1.10. Halite—NaCl and Sylvite—KCl

Clusters of sylvite and halite micro-cubes were observed in association with bitumen particles (Figure 20A). Similar sylvite micro-cubes were observed on the surface of a calcite-kaolinite "euhedra" while platey sylvite crystals were found incorporated in the calcite-kaolinite "flow".

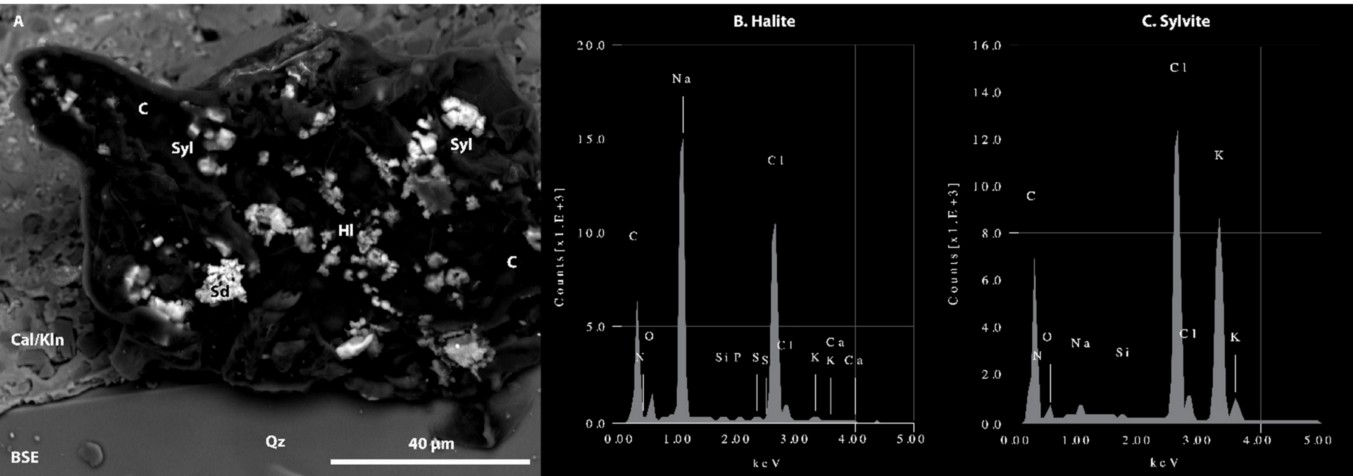

**Figure 20.** (**A**) A bitumen (C) particle featuring clusters of halite (Hl) and sylvite (Syl) micro-cubes, as well as siderite (Sd). (BSE) (**B**) Halite EDX and (**C**) Sylvite EDX from the corresponding minerals in the associated image.

### 4.1.11. Baryte—BaSO$_4$

Baryte was observed as a common inclusion in the outermost regions of the chalcedony shell and as prismatic micro-crystals associated with the calcite-kaolinite habits (Figure 9). Baryte also occurs as an irregular growth associated with siderite on top of a druzy quartz crystal.

### 4.1.12. Celestine—SrSO$_4$

Celestine was observed as an inclusion in the chalcedony shell and, in one instance, in close proximity to a baryte inclusion.

### 4.1.13. Hollandite—Ba(Mn$^{+4}_6$Mn$^{+3}_2$)O$_{16}$

Irregular growths of hollandite are observed on top of a druzy quartz crystal intergrown with siderite and associated with kaolinite (Figure 21A).

### 4.1.14. Rare Earth and Unidentified Phases

A micron-sized spherule containing lanthanum and cerium (Figure 21B) was observed protruding from a bitumen particle. We also observe several other mineral phases containing chromium, nickel, molybdenum (Figures 21C and 22), tin, copper (Figure 21D), zinc, and lead. However, due to their size and proximity to other minerals, we cannot identify the species in confidence. Many of these phases are associated with the calcite "fill" (Figure 22) or occur on top of quartz crystals similarly to siderite, baryte, and hollandite thus corresponding to the later stages of paragenesis. During our preliminary thin-section analysis we observe another REE micro-spherule and tentatively (SEM/EDX) identify zircon, rutile, and xenotime (Figure 21E–G) associated with the calcite "fill" within the chalcedony shell. Additionally, occurrences of micron sized grains containing gold and silver have been observed included in the calcedony shell (Figure 21H,I).

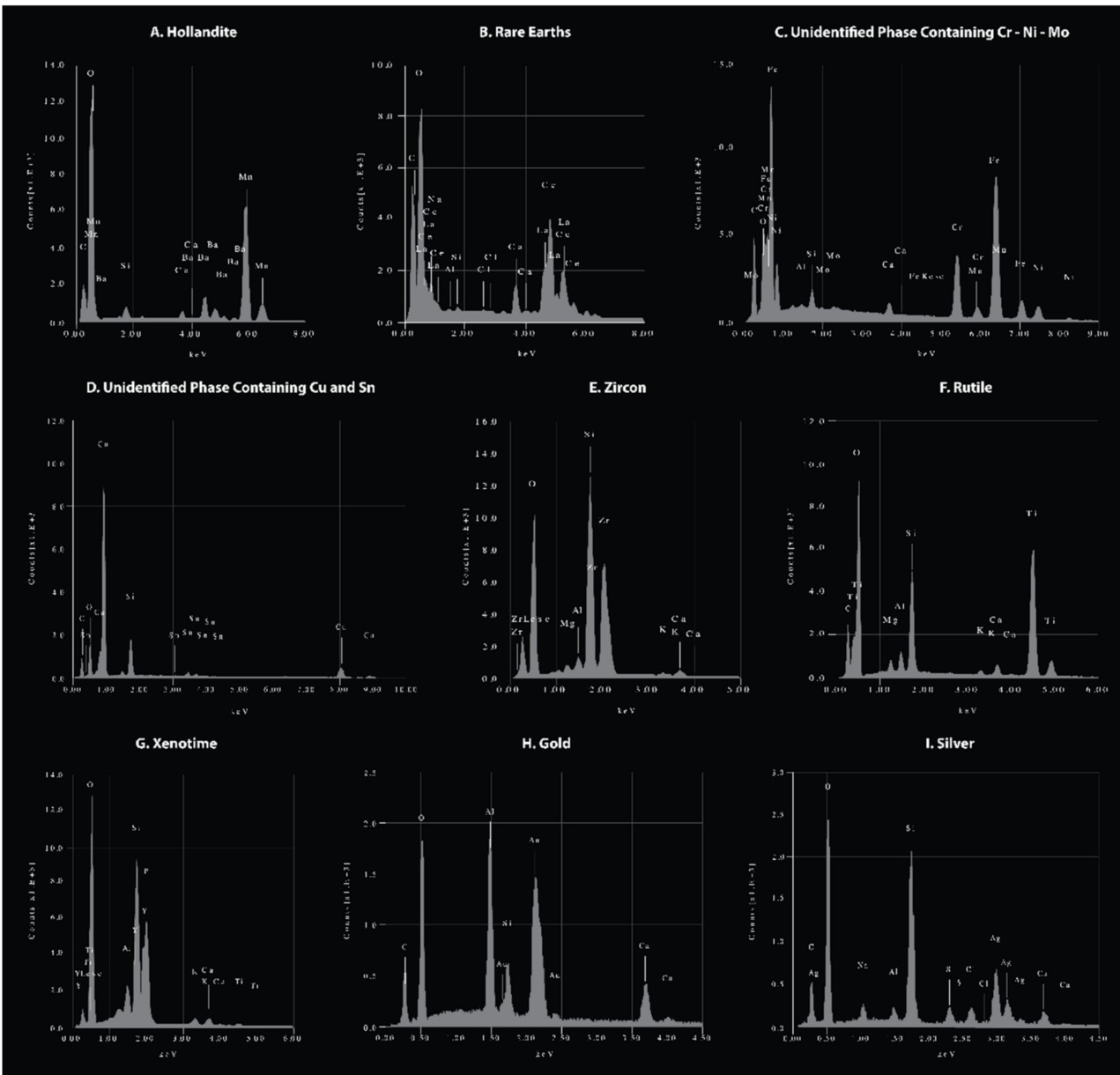

**Figure 21.** EDX Spectra of (**A**) Hollandite (**B**) REE Lanthanum and Cerium (**C**) Mineral grain containing chromium, nickel, and molybdenum observed on a dolomite crystal associated with the calcite "fill" (**D**) Mineral grain containing copper and tin observed in association with quartz. (**E**) Zircon (**F**) Rutile (**G**) Xenotime (**H**) Grain containing gold (**I**) Grain containing silver.

*4.2. Bitumen (Organic)*

Under the binocular microscope, the dark irregular particulate observed throughout the geode is identified as hydrocarbon. Bitumen in geodes is rare but has been reported in geodes from Illinois just north of the Hamilton, Illinois location where we collected the geode in this study. Cross and Zeitner [5] review the theories on the origin of the bitumen and conclude that there is no satisfactory explanation. Most of these bitumen particles we observe contain sylvite and halite micro-cubes (Figure 20) and in one instance a micron-sized REE spherule was observed protruding from a bitumen particle.

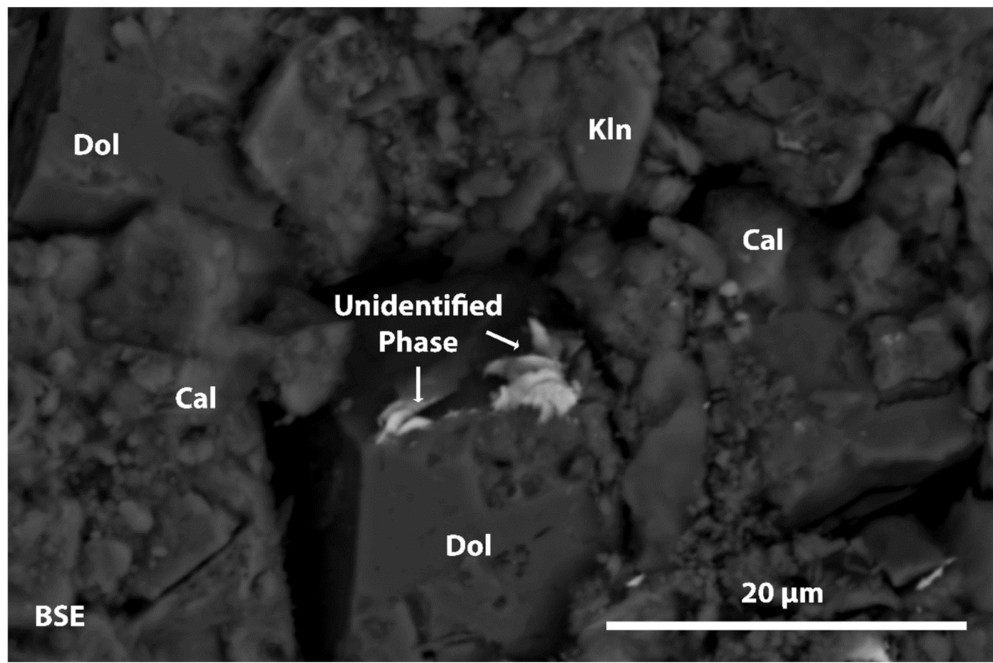

**Figure 22.** A splay of crystals containing chromium, nickel, and molybdenum (Unidentified Phase) on top of dolomite (Dol) associated with the calcite (Cal) "fill" with kaolinite (Kln). (BSE).

*4.3. Fluid Inclusions*

Thin-section analysis reveals fluid inclusions in some of the "druzy" quartz crystals (Figure 23). The fluid inclusions appear to have reflectance/iridescence under RL and exhibit air bubbles under PPL.

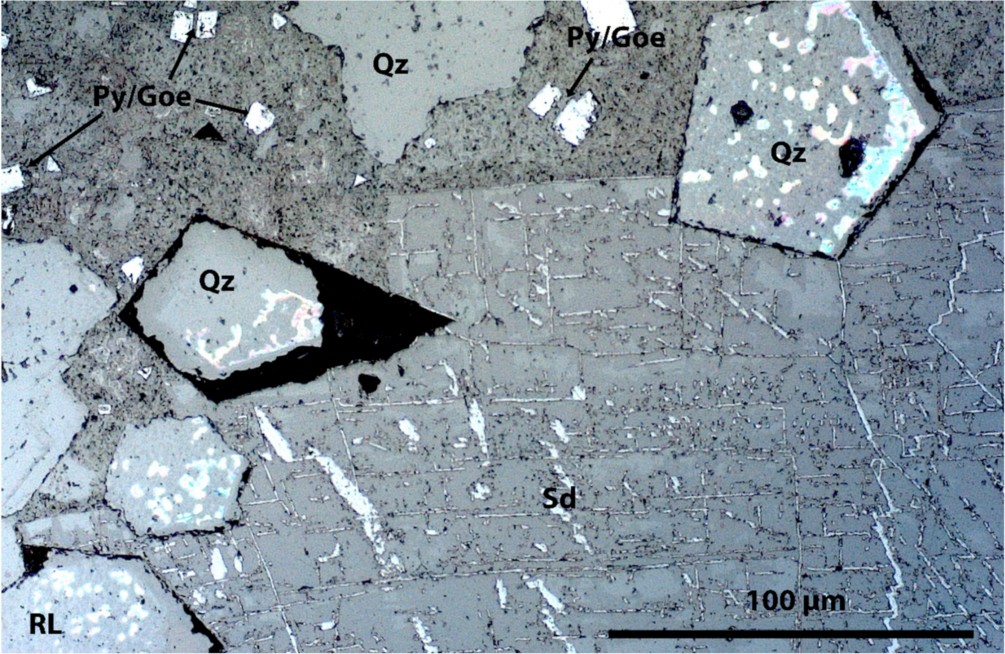

**Figure 23.** Fluid inclusions in quartz appear to exhibit reflectance/iridescence. The quartz crystals appear alongside pyrite/goethite (Py) crystals in a silica matrix alongside the siderite (Sd) 'framework". (RL).

## 5. Paragenesis

We have attempted to construct a plausible sequence of mineralization of a single geode based on the observations reported here. The paragenesis illustrates the constantly changing chemistry of the fluids percolating through the sediments and through the geode over geologic time (Figure 24). This should be considered a first pass attempt at describing the paragenesis of the specific geode of this study. We recognize at least five distinct stages of mineralization described below:

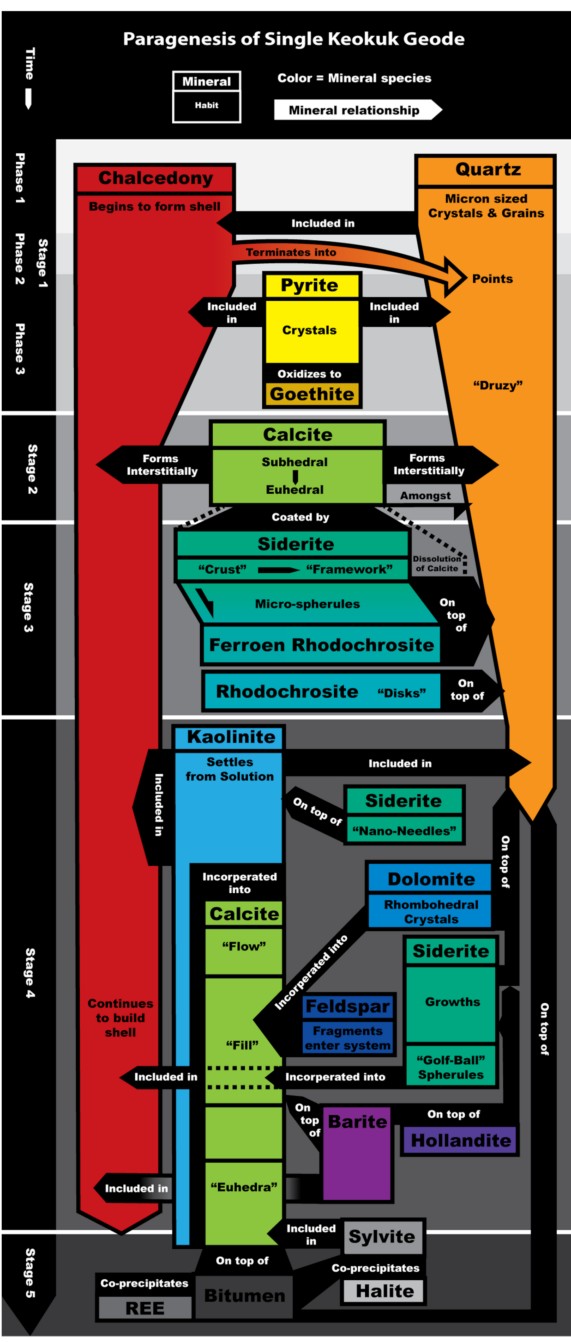

**Figure 24.** A paragenetic diagram reflecting the sequence of mineralization in this single geode based on the described mineral observations. Note: The brief examination of the thin-section revealed additional mineral relationships as well as fluid inclusions that could augment the geode paragenesis described here.

### 5.1. Stage 1: Chalcedony, Quartz, and Pyrite

Phase 1. The paragenesis of the geode begins as chalcedony forms the matrix of the shell, silica precipitates as micron-sized doubly terminated crystals and subhedral grains.

Phase 2. Terminated quartz crystals grow up to 50 microns long directly from the chalcedony while doubly terminated quartz crystals continue to precipitate. These discrete crystals continue to grow into stage 3, reaching up to 3 mm in length, and form the "druzy" of crystals observed on the inner surfaces of the shell. Chalcedony continues building the shell through stage 4, becoming host to many secondary minerals as described below.

Phase 3: Pyrite forms cubes and octahedral crystals, from 50–100 microns in size, that are incorporated as inclusions in the chalcedony shell and many of the druzy quartz crystals. When not included, the surfaces of the pyrite crystals oxidize to goethite and can be found nestled among the druzy quartz.

### 5.2. Stage 2: Calcite Crystals

Mg-bearing calcite precipitates forming interstitial subhedral crystals and larger clusters of rounded rhombohedral crystals among the druzy quartz.

### 5.3. Stage 3: Siderite-Rhodochrosite and Kaolinite

Groups of micron sized spherules ranging from siderite to ferroan rhodochrosite form directly on top of micro-quartz terminations. Similarly, irregular beveled disks of rhodochrosite form directly from larger quartz terminations and on top of discrete druzy quartz crystals.

Siderite coats the calcite rhombohedra in a ~2 micron thick crust while simultaneously precipitating in what appears to be cleavage fractures within the calcite crystals. Siderite builds along the cleavage planes forming botryoidal "walls" that incorporate larger hollow siderite spherules. Subsequent dissolution of calcite results in an open box-like structure that we refer to as the siderite "framework".

Kaolinite crystals begin to directly settle out of solution and are incorporated into the surface of some druzy quartz crystals.

### 5.4. Stage 4: Calcite with Kaolinite, Dolomite, Siderite, Hollandite, Baryte, and Celestine

Kaolinite continues to settle from solution, concentrating in different areas within the geode, and is incorporated into the habits described below.

Siderite forms in clusters of acicular nanocrystals along the edges of kaolinite crystals that have settled on top of the siderite-coated calcite crystals and the siderite framework. In areas where larger amounts of kaolinite have accumulated, siderite grows upon and in-between the individual crystals, leaving their hexagonal faces exposed.

A second generation of calcite precipitates forming 3 distinct habits related to the location within the geode and the amount of included kaolinite.

The first habit is the calcite "flow" which includes less than about 10% kaolinite and exhibits growth lines which mimic ripple marks. This habit is observed on top of the druzy quartz crystals and the siderite-coated calcite crystals that line the main cavity. When associated with the siderite framework, siderite forms in flat round growths on top of this calcite "flow".

Siderite on top of druzy quartz crystals host growths of hollandite and baryte. Prismatic micro-crystals of baryte are commonly found associated with the calcite "flow". Celestine and baryte become inclusions in the outer regions of the chalcedony shell.

A distinct phase of siderite forms a conglomerate of micron-sized spherules creating larger, around 10 microns in diameter, spherules which exhibit a golf-ball like texture. These spherules are included in the chalcedony shell and become incorporated into the calcite "fill" described below.

Dolomite forms rhombohedral crystals, between 10 and 20 microns, and are observed on top of druzy quartz crystals. Similar dolomite crystals are found incorporated into the calcite "fill" and the outermost edges of the chalcedony shell.

The second habit of calcite precipitates within void spaces between druzy quartz crystals forming a conglomerate "fill" that incorporates a range of about 20–50% kaolinite. This "fill" incorporates pyrite/goethite micro-crystals, dolomite rhombohedra, and the golf-ball-like siderite spherules. In some instances, this fill incorporates feldspar fragments.

The third habit of calcite incorporates about 50% kaolinite and forms relatively large rhombohedral crystals, up to 3 mm. These calcite-kaolinite "euhedra" forms in between and on top of druzy quartz crystals.

Chalcedony precipitation ends, completing the formation of the shell.

### 5.5. Stage 5: Bitumen, Sylvite, Halite, REE

Bitumen forms amorphous masses (averaging 50–100 microns) on top of the previous mineral assemblages including druzy quartz, the siderite framework, and the calcite "flow".

Both halite and sylvite micro-cubes form in clusters on top of bitumen particles. While only sylvite was observed incorporated on the surface of a calcite-kaolinite "euhedra" and the calcite "flow".

A micro-spherule of REE's lanthanum and cerium also forms protruding from a bitumen particle.

## 6. Discussion/Conclusions

We have identified 15 minerals in a single Keokuk geode, 5 of which are unique to this study. Our extensive optical and SEM/EDX analysis reveals an array of complex mineral assemblages, intergrowths, and inclusions that help chronologically link multiple phases of paragenesis occurring in different locations within the geode. Additionally, morphology and intricate microstructures provide a window into the extreme complexity of mineral crystallization. The majority of micro-minerals we have observed correspond with the later stages of geode paragenesis, thus providing a detailed record of the secondary mineralization processes which occurred over thousands to millions of years. We did briefly examine a second, similar appearing, geode from the collection and found that its mineralogy mirrored that of the geode described in this report.

In addition to the minerals we have described, we have observed various unidentified phases containing REE's (Figure 21B) metals such as chromium, nickel, molybdenum (Figure 21C), copper, tin (Figures 21D and 22), zinc, and lead. Preliminary thin-section analysis (SEM/EDX) reveals the occurrence of the tentatively identified minerals zircon, rutile, and xenotime (Figure 21E–G) as well as grains containing gold and silver (Figure 21H,I) within the chalcedony shell. The presence of these potentially economically valuable minerals warrants further investigation to confirm their significance. The suggestion that geode micro-minerals could be a useful indicator of nearby manganese mineral deposits were initially proposed by Finkelman et al. [13–15] in a study of the minerals in Chihuahua, Mexico geodes. Considering Missouri is a known zinc and lead mining district, we conclude that Keokuk geode mineralogy also reflects the geochemistry of the Midwest region. Thus, we believe the study of geode micro-mineralogy may offer new insight into the transportation and deposition of heavy metals and REEs.

It is extraordinary that a single, moderate-sized geode contains such a variety of minerals and complex paragenesis (Figure 24). For a sample from a location where geodes have been intensively collected and studied for more than a century, it is surprising and exciting that there is still more to discover. It is also likely that an isotope geochemical study of the geodes could help to resolve the extraordinary complex paragenesis. Our observations of this geode have convinced us that a more comprehensive examination of geode micro-minerals could be very instructive in understanding large scale mineralogic processes.

**Author Contributions:** N.M. conducted all of the SEM/EDX analysis, wrote the initial draft, and prepared all of the illustrations. R.B.F. collected the samples, designed the project, helped interpret the data, and assisted with the writing. All authors have read and agreed to the published version of the manuscript.

**Funding:** This research received no external funding.

**Acknowledgments:** We thank Leah Thompson for initial guidance on the SEM/EDX and Thomas Reyes and William Bailey for preparing the thin sections and photomicrographs. N.M. also expresses her appreciation to the Geoscience Department at UTD for financial support to make a preliminary report at a Geological Society of America Conference. We acknowledge and thank the reviewers for their constructive feedback which has enhanced the manuscript.

**Conflicts of Interest:** The authors declare no conflict of interest.

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
