# Peer review of "The Extraordinary Variety and Complexity of Minerals in a Single Keokuk Geode from the Lower Warsaw Formation, Hamilton, Illinois, USA"

_minerals, doi:10.3390/min12070914_

Round 1

Reviewer 1 Report

Please see attached file with general and specific comments.

Reviewer 2 Report

The authors offer for publication a manuscript on the extraordinary variety and complexity of minerals in a single geode from Keokuk, Iowa - a place known as "the geode capital of the world". The high quality of the English language and the descriptive style make this work easy and pleasant to read.

Unfortunately, however, there are serious discrepancies between the volumes of research stated in the abstract and the reader's expectations about the amount and nature of the information received in the entire manuscript. Here are some of them:

1. The authors claim to describe at least 16 minerals 6, of which are new to this site. The mineral species described in the manuscript are as follows: quartz, calcite, siderite, rhodochrosite, kaolinite, pyrite, goethite, dolomite, feldspar, barite, celestine, halite, sylvite, and hollandite - a total of 14 and only 3 new to the site (the last 3 in list). Chalcedony is a variety of quartz. REE and Bitumen do not constitute mineral species. Zircon, rutile, and xenotime are only mentioned as tentatively identified by SEM/EDX. No other information is provided for them.

2. The use of energy dispersive X-ray (EDX) spectroscopy stated in the abstract presupposes the reporting of results for elemental identification and quantitative compositional information. There is no such information in the whole manuscript, although lines 335-338 suggest that the authors have it.

3. Little attention is paid to new mineral species in terms of their diagnosis (morphological characterization, chemical composition, etc.).

4. There is no explanation or at least speculation about the origin of bitumen, as well as the presence of REE-containing formations closely attached to it and the nature of their carrier (e.g. zircon, xenotime, fossil apatite, etc.).

5. Figure 18. presenting the paragenetic diagram reflecting the sequence of mineralization needs more explanation in the text about the manifestation of two generations of calcite and two, and perhaps three generations of siderite, reflecting the compositions of the incoming solutions in the course of mineral formation. (see more in suggestions and recommendations).

Other minor remarks:

6. The abbreviations used (of research methods and mineral names) are better to be entered in the text (in figure captions) in the order of their appearance than at the end of the manuscript.

7. Usually, a section (subsection) Materials and methods is introduced, in which, in addition to a description of the samples, brief information is given about the equipment used, experimental conditions and methodological techniques and approaches (electron microscope, binocular microscope, etc.). Full explanations of abbreviations (SEM/EDX, BSE, XPL, etc) are also entered there.

8. Line 140: At present stilpnosiderite is considered to be a variety of limonite which on its side is defined as unidentified massive hydroxides and oxides of iron, with no visible crystals, and a yellow-brown streak. 'Limonite' is most commonly the mineral species goethite, but can also consist of varying proportions of lepidocrocite, hisingerite, pitticite, jarosite group species, maghemite, hematite, etc. (mindat.org).

9. Lines 322-324 do not belong to this part of the manuscript. These are neither discussions nor conclusions.

10. Lines 349-351 do not belong to this part of the manuscript. Fig 19. should be introduced before section 5. Discussion / Conclusions.

11. Line 3 Geodearticle should be corrected

12. Figure 5. The siderite debris needs appropriate designation m/b abbreviation too.

13. Lines 243 and 244 Omit this sentence.

14. Line 326 SEM/EDAX keep the previous style of abbreviation SEM/EDX.

Suggestions and Recommendations

Suggestions and Recommendations

15. A figure presenting the (geological) map of the region and the locality of sampling would be helpful to the reader as well as photograph(s) illustrating outcrop(s) at the type locality.

16. Powder X-ray diffraction analysis can be useful for phase diagnosis and characterization of some of the minerals. It can be used to distinguish calcite varieties containing magnesium from dolomite samples for example and others.

17. The informativeness of Figure 18 can be enhanced by graphically supplemented qualitative / semi-quantitative data for a change in the composition of the penetrating solutions responsible for the mineral formation and its sequence.

Round 2

Reviewer 1 Report

Line 30, please correct spelling of "mineralogy"

Line 75, delete the word "size", just "probe current of 50-65 pA."

Line 124, delete the duplicate sentence "The arrow indicates...."

Line 141, correct the spelling of "crust"

Line 241, Please check the abbreviation used on the figure.  It looks like your figure is labeled for chalcedony (Chc) and kaolinite, not calcite, as indicated in the text and figure 18 caption.

Reviewer 2 Report

I can see that the revised version of Manuscript ID: minerals-1788350 meets most of my remarks and suggestions stated in the first report. However, in my opinion the manuscript needs some more amendments before final decision for publication as listed below:

1. Indeed the applied EDX spectra provide qualitative data for the mineral phase identification especially for the new for this locality species, but some of these rise questions as follows:

1.1. Using carbon-coated samples for phase identification of minerals containing CO3 –groups in their compositions may be misleading in the precise determination of their type e.g. siderite vs rhodochrosite. Then, it is more appropriate to use terms like a member (members) of the siderite-rhodochrosite series. The permanent presence of Ca (calcium) in the spectra should also be taken into consideration since together with calcite the three minerals belong to the calcite group.

1.2. Feldspar is a group of minerals. When it is not possible to precisely determine the species it is better to use the term “a member of the feldspar group”.

2. Celestine, barite and kaolinite (although established previously) are not at all characterized chemically and especially for celestine the information is quite scanty and not evidenced.

3. Certain information stated already in the abstract and rising questions and expectations in the readers remains unanswered even speculatively:

3.1. There is no explanation or at least speculation about the origin of bitumen, as well as the presence of REE-containing formations closely attached to it and the nature of their carrier (e.g. zircon, xenotime, fossil apatite, etc.)

3.2. Zircon, rutile, and xenotime are only mentioned as tentatively identified by SEM/EDX. No other information is provided for them although. One of the authors states in the comments following the new subsection 4.14 Rare Earth and Unidentified Phases:  “Added these in results to corrospond with conclusion and abstract ( I can add EDX too)”….I strongly recommend to add these results for the three above written phases.

3.3. The same refers for the Mineral grain containing chromium, nickel, and molybdenum observed on a dolomite crystal associated with the calcite “fill”.

3.4. Gold and silver have disappeared in the revised version as compared with the first one.

This work still keeps its descriptive (and very enthusiastic concerning the “Extraordinary Variety and Complexity of Minerals”) character. Since the word “preliminary” is met four times in the manuscript and because it is difficult to make any general conclusion about e.g. mineral formation sequence concerning the Keokuk locality as a whole using only a single geode investigation, and many questions may remain unanswered I strongly recommend the use of terms such as “preliminary” and “a case study” in the title.
